# Machine learning the microscopic form of nematic order in twisted double-bilayer graphene

João Augusto Sobral [1,2] ✉, Stefan Obernauer[2], Simon Turkel [3], Abhay N. Pasupathy [3,4] & Mathias S. Scheurer [1,2]

Modern scanning probe techniques, such as scanning tunneling microscopy, provide access to a large amount of data encoding the underlying physics of quantum matter. In this work, we show how convolutional neural networks can be used to learn effective theoretical models from scanning tunneling microscopy data on correlated moiré superlattices. Moiré systems are particularly well suited for this task as their increased lattice constant provides access to intra-unit-cell physics, while their tunability allows for the collection of high-dimensional data sets from a single sample. Using electronic nematic order in twisted double-bilayer graphene as an example, we show that incorporating correlations between the local density of states at different energies allows convolutional neural networks not only to learn the microscopic nematic order parameter, but also to distinguish it from heterostrain. These results demonstrate that neural networks are a powerful method for investigating the microscopic details of correlated phenomena in moiré systems and beyond.

Driven by the impressive improvements in machine learning (ML) in the last couple of years, exploring its potential for quantum many-body physics has recently become the subject of intense research[1,2]. For instance, ML provides powerful tools to solve inverse problems that occur frequently in physics[3–6]: given a model, it is often straightforward with conventional many-body techniques to compute observables that can be measured experimentally, whereas the often needed inverse problem of extracting the model and underlying microscopic physics from observations is much more challenging and typically even formally ill-defined. A second example of a large class of applications of ML in physics is ML-assisted analysis of experiments, in particular of those yielding image-like data like scanning tunneling microscopy (STM)[7–10], photoemission[11], and others[12–18].

In the context of applying ML algorithms to data from imaging techniques like STM, van der Waals moiré superlattices[19,20] are particularly promising for three reasons: (i) they display a huge variety of correlated quantum-many-body phenomena, such as interaction-induced insulating phases[21], magnetism[22], superconductivity[23],

electronic nematic order[24–27], which can also coexist microscopically[27,28]. Despite intense research on these phenomena over several decades, e.g., in the pnictides or cuprates, their origin and relations are still the subject of ongoing debates. However, compared to these microscopic crystalline quantum materials, moiré superlattices are (ii) highly tunable; for instance, the density of carriers can be varied within a single sample just by applying a gate voltage (as opposed to chemical doping) and even the interactions can be tuned[29]. This allows producing large data sets of measurements on a single sample, containing a lot of information on microscopic physics. This aspect, which is crucial for data-driven approaches, is further enhanced by (iii) the large moiré unit cells of these systems compared to that of microscopic crystals, increasing the relative spatial resolution of scanning probe techniques significantly. This enables experiments to probe the structure of the wave functions within the unit cell and thus provides access to microscopic physics compared to conventional quantum materials. For instance, in the extreme limit of only one degree of freedom (Wannier state or pixel) per unit cell, the broken rotational symmetry of the

[1]Institute for Theoretical Physics III, University of Stuttgart, 70550 Stuttgart, Germany. [2]Institute for Theoretical Physics, University of Innsbruck, A-6020 Innsbruck, Austria. [3]Department of Physics, Columbia University, 10027 New York, NY, USA. [4]Condensed Matter Physics and Materials Science Division, Brookhaven National Laboratory, 11973 Upton, NY, USA. ✉e-mail: joao.sobral@itp3.uni-stuttgart.de

electron liquid—the defining property of electronic nematic order[30,31]—is not visible as a consequence of translational symmetry and thus requires a careful analysis of the behavior around impurities[32].

In this work, we explore these advantages of moiré superlattices for extracting or learning effective field-theoretical descriptions of their correlated many-body physics from STM data. This can be viewed as an inverse problem and is also conceptually related to the goal of Hamiltonian learning in quantum simulation[33–38], albeit in rather different regimes and based on different measurement schemes. As a concrete example, we use electronic nematic order in twisted double-bilayer graphene (TDBG)[39–45]. This moiré system consists of two AB-stacked bilayers of graphene that are twisted against each other; as one can see in Fig. 1a, it exhibits the point group $D_3$, generated by threefold rotation $C_3$ along the out-of-plane $z$-axis and twofold rotation $C_{2x}$ along the in-plane $x$-axis. Evidence of electronic nematic order has been observed in previous STM experiments[42,46] which clearly exhibit stripe-like features breaking the $C_3$ symmetry spontaneously for certain electron concentrations. While simple limiting cases have been compared with the data in Samajdar et al.[46], there is no systematic analysis of the microscopic form of nematicity in the system. To fill this gap, we consider the more general case in which all leading terms on the graphene and moiré scale describing nematic order in a continuum-model description of TDBG[47] are included. In addition, as it is common in graphene moiré systems[24–26,42,48], we also allow for finite strain. The

Hamiltonian defining the changes in TDBG resulting from nematic order and strain depends on a set of parameters $\beta$, which we reconstruct from STM data using convolutional neural networks (CNN) in a supervised learning procedure. As such, our study differs significantly from recent works, which focused on detecting the presence or absence of nematic order[32] or performed a phenomenological data analysis of STM measurements[49] with ML, rather than extracting the underlying microscopic physics as we do here.

## Results

### Nematic order in TDBG

The non-interacting band structure of TDBG features two moiré minibands per spin and valley close to charge neutrality, where a variety of correlation-driven phenomena can emerge[39–45]. In Fig. 1b, these minibands are denoted as valence (VFB) and conduction flat bands (CFB). The band structure shown is obtained from continuum-model calculations close to half-filling of the CFB (band filling $v = 0.475$), where electronic nematic order was observed to be the strongest[42], see Supplementary Note 1 for more details. STM experiments probe the band structure and wave functions of a system by providing direct access to the spatial and energy dependence of the local density of states (LDOS). Most commonly, the LDOS is studied either for a fixed position $\boldsymbol{r}_0$ over a range of different energies, $\mathcal{D}_{\boldsymbol{r}_0}(\omega)$, or for a fixed energy $\omega_0$ covering a spatial region of the system, $\mathcal{D}_{\omega_0}(\boldsymbol{r})$.

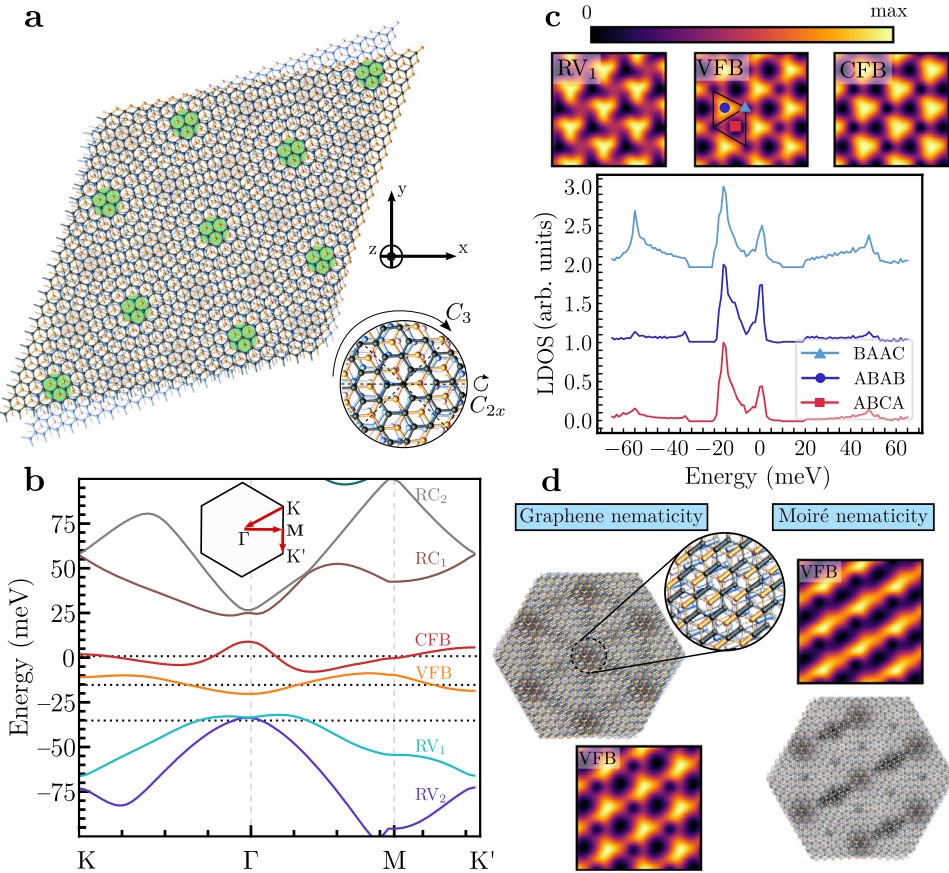

**Fig. 1 | TDBG, LDOS maps, and nematicity. a** Representation in real space of the TDBG heterostructure. Green highlighted domains emphasize the emerging moiré pattern due to the combination of two AB-stacks of graphene bilayers with a relative twist angle, which in this case is given by $\theta = 7.24°$. $C_3$ and $C_{2x}$ describe threefold and twofold rotations along the $z$- and $x$-axes, as illustrated in the small coordinate system. **b** Band structure for $\theta = 1.05°$ along highly symmetrical points from the moiré Brillouin zone (inset). Solid lines represent conduction and valence flat bands (CFB/VFB) as well as remote bands (R). The chemical potential corresponds to roughly a half-filling fraction ($v = 0.475$) of the CFB. **c** LDOS for three fixed energies

(black dotted horizontal lines in **b**) as a function of position, and for varying energy at fixed high-symmetry positions in the moiré unit cell (black rhombus). The $\mathcal{D}_{\omega_0}(\boldsymbol{r})$ map intensities are always normalized accordingly to the corresponding colorbar. The $\mathcal{D}_{\boldsymbol{r}_0}(\omega)$ map is vertically shifted for better visual comparison. The solid lines are taken from the $\boldsymbol{r}_0 = $ (BAAC, ABAB, ABCA) stacking positions in $\mathcal{D}_{\boldsymbol{r}_0}(\omega)$ maps.
**d** Schematic real-space illustration of two limiting cases of graphene and moiré nematicity, along with two samples of LDOS plots for fixed energy in the VFB; both show clear $C_3$ symmetry breaking.

The behavior of $\mathcal{D}_{\omega_0}(\mathbf{r})$ and $\mathcal{D}_{\mathbf{r}_0}(\omega)$ following from the continuum model for TDBG for three different energies and high-symmetry positions in the moiré unit cell is shown in Fig. 1c. The $C_3$ rotational and translational symmetry of the moiré lattice can be clearly seen in $\mathcal{D}_{\omega_0}(\mathbf{r})$. Meanwhile, $C_{2x}$ is broken, albeit weakly, as a consequence of the electric field required to control the electron filling to be close to the middle of the CFB in an open-faced STM sample geometry[42].

In graphene moiré systems, there are two fundamentally distinct sources of $C_3$ symmetry breaking−strain and electronic nematic order. Postponing the discussion of the former below, electronic nematic order[30,31] refers to the spontaneous rotational symmetry breaking as a result of electronic correlations. While recent works also indicate the possibility of nematic charge-density wave states in TDBG[43,50], where moiré translational symmetry is simultaneously broken, we here focus on translationally symmetric nematic order since the STM data of Rubio-Verdú et al.[42] preserves moiré translations. The underlying nematic order parameter we study is a time-reversal- and moiré-translation-invariant vector $\boldsymbol{\Phi} = \Phi\hat{\boldsymbol{\Phi}}_\varphi$, $\hat{\boldsymbol{\Phi}}_\varphi = (\cos 2\varphi, \sin 2\varphi)$, transforming under the irreducible representation $E$ of $D_3$ (or of $C_3$, taking into account the weak $C_{2x}$ breaking); $\Phi$ and $\varphi$ stand for the intensity and orientation of the nematic director, respectively. The microscopic form of nematicity can be modeled by a coupling of $\boldsymbol{\Phi}$ to a fermionic bilinear and reads in its most general form in a continuum-model description as[46]

$$\mathcal{H}_{\boldsymbol{\Phi}} = \int_{\mathbf{r}}\int_{\Delta\mathbf{r}} \boldsymbol{\Phi} \cdot \boldsymbol{\phi}_{\sigma,\ell,s,\eta;\sigma',\ell',s',\eta'}(\mathbf{r},\Delta\mathbf{r}) \\ \times c^\dagger_{\sigma,\ell,s,\eta}(\mathbf{r}+\Delta\mathbf{r})c_{\sigma',\ell',s',\eta'}(\mathbf{r}) + \text{H.c.},$$ 

(1)

where $c^\dagger$ and $c$ are the electronic creation and annihilation operators. This general form encompasses couplings between the two sublattices $s = A, B$ of the microscopic graphene sheets, the four graphene layers $\ell = 1, ..., 4$, the valley $\eta = \pm$ and spin $\sigma = \uparrow, \downarrow$ degrees of freedom in the tensorial form factor $\boldsymbol{\phi}_{\sigma,\ell,s,\eta;\sigma',\ell',s',\eta'}(\mathbf{r},\Delta\mathbf{r})$; its two components are required to transform in the same way as $\boldsymbol{\Phi}$ under all symmetries of the system. In the following, we will take $\boldsymbol{\phi}$ to be trivial in the spin and diagonal in the valley indices, $\boldsymbol{\phi}_{\sigma,\ell,s,\eta;\sigma',\ell',s',\eta'} = \delta_{\sigma,\sigma'}\delta_{\eta,\eta'}\boldsymbol{\phi}_{\ell,s;\ell',s'}(\eta)$. This is motivated by the weak spin-orbit coupling in graphene[51,52] and the lack of indications of interaction-induced spin-orbit coupling, which is also strongly constrained[53]. Furthermore, the intervalley-coherent nematicity is known to lead to stronger effects on the remote bands[46] that were not observed experimentally[42].

Since we are working with a continuum theory, the space of possible couplings $\boldsymbol{\phi}$ in Equation (1) is technically infinite-dimensional. As such, a complete reconstruction of $\boldsymbol{\phi}$ from experimental data is impossible given the finite resolution and energy range of the available data. On top of this, it is not required either as we are primarily interested in understanding the low-energy behavior of the system. In the spirit of gradient expansions commonly used in continuum low-energy field theories, we will therefore only keep the leading terms in $\boldsymbol{\Phi}$. There is, however, a subtlety associated with the presence of an additional moiré length scale. We will therefore have to consider two basic classes of nematic orders, referred to as graphene (GN) and moiré (MN) nematicity[42,46].

In the case of MN, nematic order is associated with the moiré scale, i.e., we choose $\Delta\mathbf{r} = \mathbf{R}_{m_1,m_2} = m_1\mathbf{L}_1^M + m_2\mathbf{L}_2^M$ in Equation (1), $m_j \in \mathbb{Z}$, with moiré lattice vectors $\mathbf{L}_j^M$, to represent the non-trivial transformation behavior of $\boldsymbol{\phi}$ under $C_3$. We can thus take it to be diagonal in the remaining internal indices, yielding

$$\mathcal{H}_{\boldsymbol{\Phi}}^{\text{MN}} = \frac{1}{2}\Phi_{\text{MN}}\int_{\mathbf{r}}\sum_{m_1,m_2\in\mathbb{Z}}\hat{\boldsymbol{\Phi}}_{\varphi_{\text{MN}}} \cdot \boldsymbol{\phi}_{m_1,m_2}(\mathbf{r}) \\ \times c^\dagger_\alpha(\mathbf{r}+\mathbf{R}_{m_1,m_2})c_\alpha(\mathbf{r}) + \text{H.c.},$$

(2)

with multi-index $\alpha = (\sigma, \ell, s, \eta)$. We further focus on the lowest moiré-lattice harmonic by setting $\phi_{m_1,m_2}(\mathbf{r}) = \phi_{m_1,m_2}$ and only keeping the

terms with the shortest possible $\mathbf{R}_{m_1,m_2}$. Intuitively, MN order can be thought of as a distortion of the effective inter-moiré-unit-cell hopping matrix elements, as illustrated schematically in the lower right panel of Fig. 1d.

Conversely, GN acts as a local order parameter, $\Delta\mathbf{r} = 0$ in Equation (1), without any explicit reference to the moiré scale,

$$\mathcal{H}_{\boldsymbol{\Phi}}^{\text{GN}} = \Phi_{\text{GN}}\int_{\mathbf{r}}\hat{\boldsymbol{\Phi}}_{\varphi_{\text{GN}}} \cdot \boldsymbol{\phi}_{\ell,s;\ell',s'}(\eta;\mathbf{r})c^\dagger_{\ell,s}(\mathbf{r})c_{\ell',s'}(\mathbf{r}).$$

(3)

Here, the correct transformation properties of $\boldsymbol{\phi}$ result from its structure in the internal indices. Focusing on the local intra-layer contributions and the leading (constant) basis function, the most general form reads as

$$\boldsymbol{\phi}_{\ell,s;\ell',s'}(\eta;\mathbf{r}) = \delta_{\ell,\ell'}\psi_\ell \begin{pmatrix} (e^{i\alpha_\ell\eta\rho_z}\rho_x)_{ss'} \\ \eta(e^{i\alpha_\ell\eta\rho_z}\rho_y)_{ss'} \end{pmatrix},$$

(4)

where Pauli matrices in sublattice space are represented by $\rho_j$; $\alpha_\ell$ and $\psi_\ell$ are real-valued parameters. As shown schematically in the upper left panel of Fig. 1d, one can think of GN as the nematic distortion of the bonds of the individual graphene layers in a way that preserves the graphene translational symmetry.

We emphasize that GN and MN should not be viewed as distinct phases; they break the same symmetries and as such in general mix. We thus take $\mathcal{H}_{\boldsymbol{\Phi}}^{\text{MN}} + \mathcal{H}_{\boldsymbol{\Phi}}^{\text{GN}}$ to describe nematicity in TDBG in the following, which depends on the set of parameters $\beta = \{\alpha_\ell, \psi_\ell, \Phi_{\text{MN}}, \Phi_{\text{GN}}, \varphi_{\text{MN}}, \varphi_{\text{GN}}\}$. The computation of the LDOS for a specific set of parameters can be done straightforwardly from the continuum model. The resulting spatial dependence of the LDOS, $\mathcal{D}_{\omega_0}(\mathbf{r})$, is also shown in Fig. 1d for two different values of $\beta$. As opposed to the plots without nematic order, $C_3$ is now broken, leading to stripes in the VFB, while translational symmetry is still preserved. The inverse problem−inferring the value of the parameters $\beta$ from a given LDOS pattern−is a much more challenging task. Our goal in the following sections will be to use ML, in particular, CNNs to learn the set $\beta$ directly from LDOS images.

## Data sets and learning stage

Using CNNs to solve this inverse problem can be interpreted as a supervised learning task[2], i.e., a regression-like procedure using synthetic LDOS data labeled by their respective value of nematicity parameters $\beta$. More specifically, our CNNs take as inputs $65 \times 65$ pixels of LDOS images and apply consecutive transformations (represented by a set of weights between each layer) in order to extract meaningful correlations that represent the set $\beta$. One example of the CNN image inputs is shown in Fig. 2a. The complete data set consists of 12,000 images which are divided into training (60%), validation (20%), and test (20%) subgroups. Each image is generated for a randomly sampled set of nematic parameters $\beta$ and the intensities in the LDOS are modified with the addition of Gaussian noise (see Supplementary Note 1). The motivation for noise is twofold: to avoid overfitting[54] and to test the stability against and performance of the procedure with noise, which is inevitably present in experimental data. For a detailed description of the CNN architecture, see the Methods section and Fig. 2a.

The learning procedure is then defined by the minimization of the loss function with respect to the CNN's weights in a backward propagation procedure[55]. The loss function can be represented as the mean-squared error (MSE), which is defined as the difference between the true and expected set of parameters $\beta$ in $\text{MSE} = \sum_j^N \left(\beta_j^{\text{true}} - \beta_j^{\text{predicted}}\right)^2/N$, with $N$ representing the number of samples in the training or validation data sets. Finally, we consider the adaptive moment estimation (ADAM) for the minimization of the loss function, with a learning rate of 0.001 and batch size equal to 64[56]. After the completion of the training stage, the algorithm is ready to be

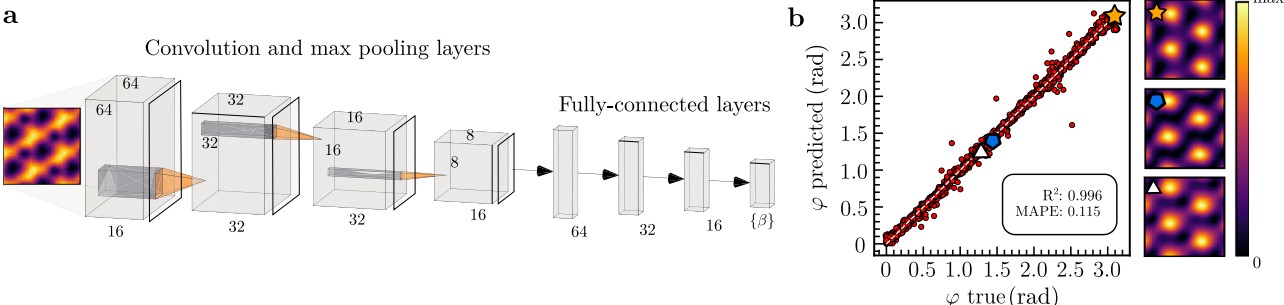

**Fig. 2 | CNN architecture and nematic director prediction. a** Schematic figure of the CNN architecture used with only one $\mathcal{D}_{\omega_0}(\boldsymbol{r})$ input channel at an energy $\omega_0$ in the VFB, see the Methods section for details on the architecture and the main text for information about the data sets. In the last linear layer, $\beta$ represents the set of learnable parameters. **b** Comparison between true and predicted nematic director angles $\varphi$. The white dashed line serves to guide the eye. $R$-squared ($R^2$) and mean absolute percentage error (MAPE) metrics are shown in the inset. Details on how these metrics are calculated can be seen in the Methods section. Three samples of $\mathcal{D}_{\omega_0}(\boldsymbol{r})$ (star, pentagon, and triangle) are displayed to emphasize that the relation between the LDOS and $\varphi$ is highly non-trivial as a result of the presence of different forms of nematicity.

deployed to previously unseen data, returning as outputs the parameters $\beta^{\text{predicted}}$.

## Orientation of the nematic director

As a first investigation, we consider the task of predicting the orientation $\varphi$ of the nematic director from $\mathcal{D}_{\omega_0}(\boldsymbol{r})$ images at a single energy in the VFB ($\omega_0 = -15$ meV, see Fig. 1b). For this, we consider a data set with randomly generated MN and GN intensities $\Phi_{\text{MN}}, \Phi_{\text{GN}} \in [0.001, 0.1]$ eV, and $\varphi_{\text{MN}} = \varphi_{\text{GN}} = \varphi \in [0, \pi]$. Furthermore, $\psi_l = 1$ and $\alpha_l = 0$ for all layers. The relation between the shape of the LDOS at single energy $\mathcal{D}_{\omega_0}(\boldsymbol{r})$ and $\varphi$ is highly non-trivial for two reasons: even for a given form of nematicity, changing $\varphi$ generically not just merely rotates the LDOS pattern, due to the lattice, but leads to complex distortions of its structure. Additionally, by sampling $\mathcal{H}_{\boldsymbol{\Phi}}^{\text{MN}} + \mathcal{H}_{\boldsymbol{\Phi}}^{\text{GN}}$, even if the same bond direction is favored over the $C_3$-related ones in the LDOS pattern of two samples, the underlying $\varphi$ can be rather different. As can be seen in the three sample LDOS plots in Fig. 2b with different values of $\varphi$, the correspondence between $\varphi$ and $\mathcal{D}_{\omega_0}(\boldsymbol{r})$ is complex and not apparent to the human eye.

Using the angles $\varphi$ as labels to the data is the most straightforward choice, but leads to inaccurate predictions around 0 and $\pi$ due to the periodicity in the definition of the nematic order parameter, $\hat{\boldsymbol{\Phi}}_{\varphi} = (\cos 2\varphi, \sin 2\varphi) = \hat{\boldsymbol{\Phi}}_{\varphi+\pi}$. To circumvent this feature, we use the two-component label $\hat{\boldsymbol{\Phi}}_{\varphi}$ instead of $\varphi$ in the training process and then fold the network's prediction back to $\varphi$ with the arctan2 function[57]. The results, shown in Fig. 2b, are consistent with the true labels, including at the boundaries of $\varphi$'s domain. This shows that even when the precise nature of nematicity (predominantly MN or GN or an admixture of the two) is not known, the director orientation $\varphi$ can be accurately predicted with our CNN setup from $\mathcal{D}_{\omega_0}(\boldsymbol{r})$ at a single energy. We have checked that the few outliers in Fig. 2b are directly related to small nematic intensities, where $\varphi$ has virtually no impact on the LDOS and is, thus, impossible to predict.

## Form of nematicity

After successfully learning the director orientation $\varphi$ in the presence of different nematicities, we proceed into investigating the finer details of these couplings by learning the parameters $\beta = \{\Phi_{\text{MN}}, \Phi_{\text{GN}}, \alpha_l\}$ defined in Equations ((2)–(4)). To this end, we consider $\psi_l = 1$ and $\alpha_l = \alpha$ for all layers. For concreteness, we set $\varphi_{\text{MN}} = \varphi_{\text{GN}} = \varphi = 2\pi/3$, which is one of the possible discrete orientations ($\varphi_{\text{MN}} = \varphi_{\text{GN}} = 2\pi/3, \pi/6$ and symmetry related) of the nematic director in the presence of $C_{2x}$. The data set now consists of randomly generated MN and GN intensities $\Phi_{\text{MN}}, \Phi_{\text{GN}} \in [0.001, 0.1]$ eV, and $\alpha \in [0, \pi]$. The intensity values are chosen such that the stripes in the VFB resemble the experimental

results[42]. As with $\varphi$, instead of learning the angular variable $\alpha$ directly, the arctan2 mapping is also applied.

Using only the LDOS at a single energy (i.e., one $\mathcal{D}_{\omega_0}(\boldsymbol{r})$ channel) in the ML architecture for this task does not produce accurate predictions. Additionally, both hyperparameter optimization and architecture modifications did not lead to any significant improvement, implying that nematic order impacts the electronic structure in complex ways that cascade across energy scales. In fact, this is also intuitively clear since, for example, the samples marked by a star and pentagon in Fig. 3a have fundamentally different nematic couplings and yet exhibit visually similar $\mathcal{D}_{\omega_0}(\boldsymbol{r})$ images at the VFB energy.

In experiments, one can typically obtain single-point spectra [$\mathcal{D}_{\boldsymbol{r}_0}(\omega)$] and real-space LDOS images at fixed energies [$\mathcal{D}_{\omega_0}(\boldsymbol{r})$]. We can therefore include additional input channels corresponding to $\mathcal{D}_{\omega_0}(\boldsymbol{r})$ and $\mathcal{D}_{\boldsymbol{r}_0}(\omega)$ for different energies $\omega_0$ and points $\boldsymbol{r}_0$, respectively. In the second case, the individual point spectra are transformed to scaleogram images for consistency with the input data for CNNs[5,58], see upper left inset in Fig. 3a and Supplementary Fig. 1. The new architecture is then formed by four channels with $\mathcal{D}_{\omega_0}(\boldsymbol{r})$ inputs at fixed energies $\omega_0 = (-35, -15, 1, 23)$ meV within the flat and remote bands, such that they resemble visually the corresponding ones in the experimental data of Rubio-Verdú et al.[42], and three channels for $\mathcal{D}_{\boldsymbol{r}_0}(\omega)$ scaleogram inputs at stacking positions $\boldsymbol{r}_0 = (BAAC, ABAB, ABCA)$, cf. Fig. 1c. Each channel is passed through parallel Conv-Batch-MaxPool layers as in Fig. 2a, but instead of flattening each channel separately, they are concatenated to a Dense-Dropout stage before the last layer (Fig. 3a).

In Fig. 3b–d, predictions on the test data set are represented for (b) $\alpha$, and (c) the moiré and (d) graphene nematic intensities; as can be seen, very good agreement is found between the reconstructed and true parameters. The outliers in $\alpha$ are related to small $\Phi_{\text{GN}}$ (brighter colors). From Equations (3) and (4), it is clear that for small $\Phi_{\text{GN}}$, minimal changes will be induced in the LDOS, irrespective of the true value of the phase governed by $\alpha$. This is a similar behavior to what was observed for outliers in the nematic director prediction. The results of Fig. 3 demonstrate that the microscopic form of nematicity can be extracted from the LDOS if significant energy dependence is included in the input data set.

## Including strain

As already alluded to above, another possible source of $C_3$ breaking is strain[48,59–61], which is believed to be a ubiquitous property of graphene moiré superlattices at small twist angles. Breaking the same symmetries as nematic order, strain can obscure the experimental identification of nematic order and their precise interplay is still under debate[24–26,62]. Experiments indicate[24–26,42,48] that the most relevant form of strain in graphene superlattices such as twisted bilayer graphene

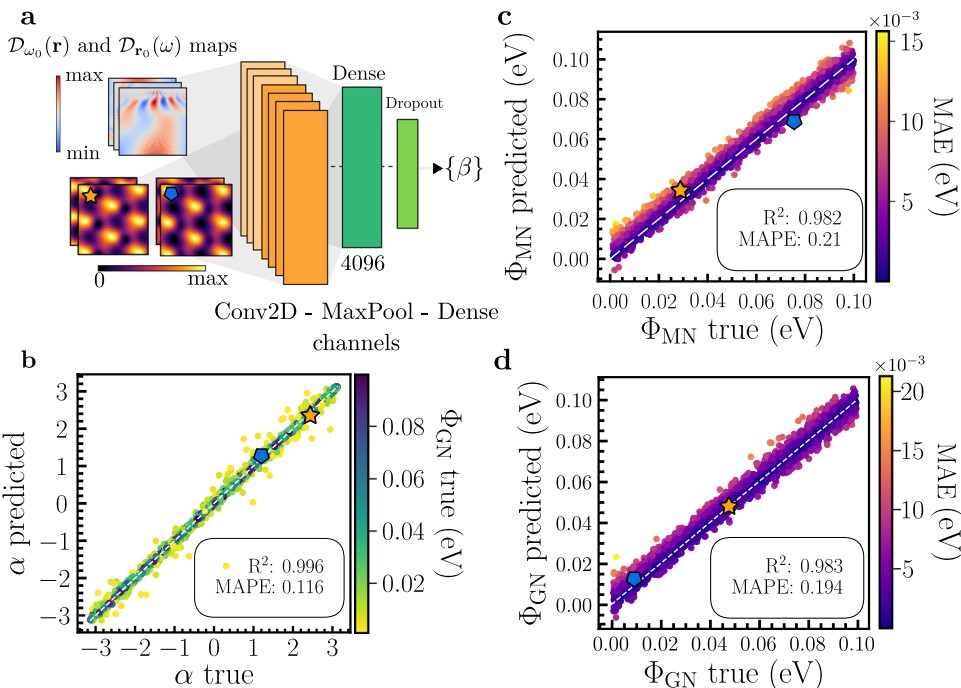

**Fig. 3 | Predicting the form of nematicity. a** CNN architecture used for learning the nematic microscopic parameters. Each orange rectangle labeled as `Conv2D-MaxPool-Dense' refers to the structure from Fig. 2a. The last Dense linear layer is now followed by a Dropout layer to prevent overfitting. The input is based on scaleograms (see Supplementary Note 1) of $\mathcal{D}_{\mathbf{r}_0}(\omega)$ in addition to the previously seen $\mathcal{D}_{\omega_0}(\mathbf{r})$ maps. Both are normalized accordingly to their corresponding color-bars. **b** Predicted versus true $\alpha$ parameter, with outliers (brighter colors) being related to small graphene nematic intensity $\Phi_{GN}$. **c, d** Predicted versus true parameters for graphene and moiré intensities, with colorbars representing the mean absolute error (MAE) in the intensities. The white dashed lines serve to guide the eye. $R$-squared ($R^2$) and mean absolute percentage error (MAPE) metrics are shown in the inset. Details on how these metrics are calculated can be seen in the Methods section. Star and hexagon symbols are examples indicating that two very different forms of nematicity can lead to very similar LDOS patterns at a single energy, making the inclusion of several channels necessary.

(TBG) or TDBG is uniaxial heterostrain. In this case, the matrices $\mathcal{E}_j$ describing the in-plane metric deformation of the coordinates in the $j$th rotated Bernal bilayer of TDBG are of the form

$$\mathcal{E}_2 = -\mathcal{E}_1 = \frac{1}{2} R(\theta_\epsilon)^{-1} \begin{pmatrix} -\epsilon & 0 \\ 0 & \nu\epsilon \end{pmatrix} R(\theta_\epsilon). \quad (5)$$

Here $\nu = 0.16$ is the Poisson ratio for graphene and $R(\theta_\epsilon)$ is the $2 \times 2$ matrix describing rotations of 2D vectors by angle $\theta_\epsilon$. We see that uniaxial heterostrain is characterized by two variables, the strain intensity $\epsilon$ and the direction of strain, parameterized by the angle $\theta_\epsilon$.

In the following, we allow for the simultaneous presence of uniaxial heterostrain and nematic order, leading to two additional parameters, $\epsilon$ and $\theta_\epsilon$, in $\beta$. We will study whether our ML approach is still able to extract the microscopic form of nematicity and also learn the relative strength and direction of strain. Note that the form of nematicity is still given by Equations ((2)–(4)), with the only difference that we replace $\mathbf{L}_j^M$ in the definition of $\mathbf{R}_{m_1,m_2}$ by the strained moiré lattice vectors. The data set for this task is built with nematic intensities $\Phi_{MN}, \Phi_{GN} \in [0.001, 0.1]$ eV, with the addition of strain parameters $\epsilon \in [0, 0.8]\%$ and $\theta_\epsilon \in [0, \pi/3]$. Here, $\alpha_l = 0$, $\psi_l = 1$ and $\varphi = \varphi_{MN} = \varphi_{GN} = 2\pi/3$. The domain for the strain intensities is chosen based on typical values observed in TBG[24], and for $\theta_\epsilon$ on the periodicity of the unstrained system as $\theta_\epsilon \to \theta_\epsilon + \pi/3$[61]. The ML architecture employed in this section is the same as in the previous investigation (Fig. 3a).

In Fig. 4a–d, predictions on the test data set are shown for $\epsilon$ (a), $\theta_\epsilon$ (b), and the nematic intensities (c, d). At first sight, the result for the strain angle in Fig. 4b looks as if the procedure ceased to work since there are many data points where the true and predicted value of $\theta_\epsilon$

differ significantly. However, when indicating the true strain intensity label $\epsilon$ for each prediction, it becomes clear that the outliers are related to small values of $\epsilon$ (brighter colors). As such, this behavior is not a shortcoming of the learning procedure but actually a feature of strain: for small enough $\epsilon$ in Equation (5), the angle $\theta_\epsilon$ has no meaning. We have checked that removing the samples with small strain $\epsilon$ from the training and test data set will lead to accurate predictions of $\theta_\epsilon$ (see Supplementary Fig. 2). The stability that we find for our learning procedure in the presence of virtually vanishing $\epsilon$ is, however, important when applying it to experimental data, where the strength of strain is unknown.

Most importantly, we see in Fig. 4c, d that the nematic couplings can still be accurately predicted when varying strain is present. The MAE is equally distributed in these cases, in contrast to the strain intensity prediction. This shows that not only nematic order can be identified when strain is present, but also its internal structure and the strength of strain that is present at the same time can be resolved when using different channels consisting of both $\mathcal{D}_{\mathbf{r}_0}(\omega)$ and $\mathcal{D}_{\omega_0}(\mathbf{r})$ as inputs. This allows the networks to take into account correlations between different energies in the STM data, which in turn conveys the crucial microscopic physics, enabling the model to disambiguate between lattice and electronic effects.

## Experimental data

After demonstrating the effectiveness of CNNs on learning microscopic parameters $\{\beta_i\}$ from a synthetic (theoretical) data set $D_{th}(\beta_1, \cdots, \beta_{N_{th}})$ with $N_{th}$ samples, we now proceed into applying the trained ML architecture for predictions of the a priori unknown sets of parameters $\{\beta_i'\}$ in an experimental data set $D_{exp}(\beta_1', \cdots, \beta_{N_{exp}}')$. For concreteness, we use the same synthetic training data set as in

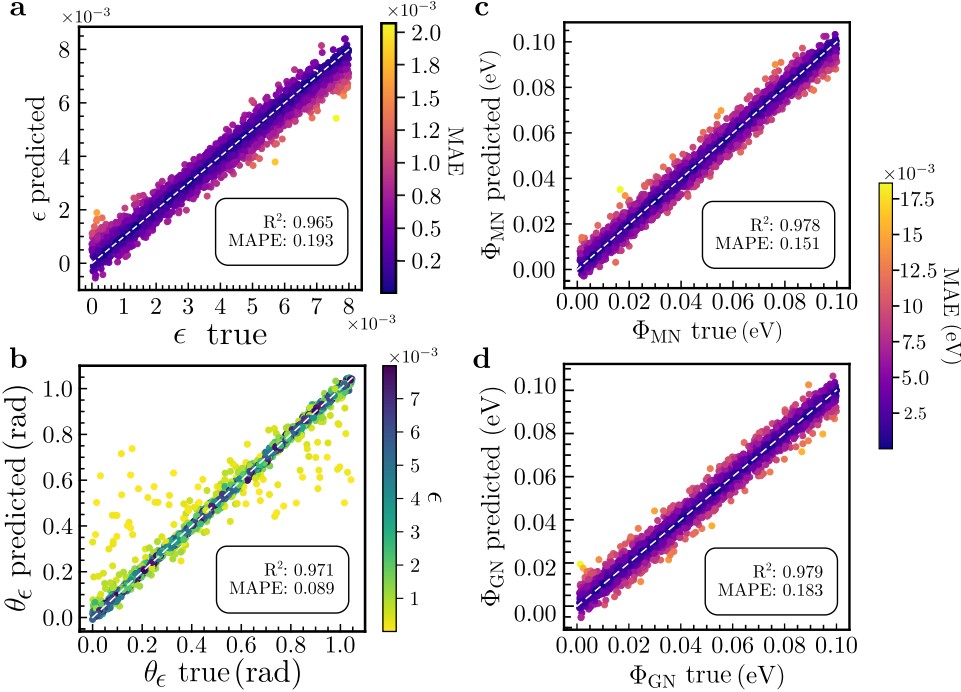

**Fig. 4 | Distinguishing strain and nematicity.** Predicted versus true values for the strain intensity $\epsilon$ (**a**) and angle $\theta_\epsilon$ (**b**). The prediction for the nematic intensities is depicted in panels **c** and **d**. The white dashed lines serve to guide the eye. $R$-squared ($R^2$) and mean absolute percentage error (MAPE) metrics are shown in the inset. Details on how these metrics are calculated can be seen in the Methods section. The CNN architecture used to produce these results is described in Fig. 3a. Similarly to the prediction of the $\alpha$ parameter in the presence of only nematicity, outliers in $\theta_\epsilon$ are related to small $\epsilon$.

Supplementary Note 2, where only the nematic and strain intensities are predicted, i.e., $\beta = \{\Phi_{MN}, \Phi_{GN}, \epsilon\}$. The data set $D_{exp}$ is constituted of both scaleograms $\mathcal{D}_{r_0}(\omega)$ and $\mathcal{D}_{\omega_0}(\mathbf{r})$ maps for different fillings of the CFB ($n_s$). More details about the preprocessing of the experimental data $D_{exp}$ can be found in the Supplementary Fig. 3.

In Fig. 5, predictions of the trained CNN for the set $\{\beta'_i\}$ show non-zero values of nematicity (a) and strain (b) for all fillings of the CFB. For $n_s \geq 0.47$ (gray region), the experimental data shows the most pronounced signatures of broken rotational symmetry to the human eye, which was previously interpreted as electronic nematic order[42,46]. Here the CNN predicts MN to dominate over GN, although both are finite (as expected by symmetry). As can be seen in Fig. 5c, the parameters predicted by the CNN nicely reproduce the key features in the experimental data, including the strong stripes in the VFB and the much weaker, albeit finite, signatures of nematicity in the other bands.

For smaller fillings, $n_s < 0.47$, the experimental data still exhibit distortions that break $C_3$, see Supplementary Fig. 4, but no clear stripe-like features appear. The CNN tries to assign different anisotropy sources to these distorted regions, but the agreement between theoretical prediction and experiment is less accurate than for larger $n_s$. It is clearly possible that, indeed, a crossover from primarily MN to GN occurs when lowering $n_s$, as predicted by the neural network, see Fig. 5a, in particular, since nematic order is also a plausible instability in non-twisted bilayer graphene[29,63]. However, we believe that additional experimental data and refined theoretical models are required to conclude whether this is really the case.

In contrast to this interplay between the nematic couplings, strain remains relatively constant for all $n_s$, and slightly decreases in Fig. 5b for $n_s \geq 0.47$ as it approaches the same order of magnitude of $\epsilon \in [0.003 - 0.1\%]$ that is expected for the experimental samples in $D_{exp}$[42]. We note that at low fillings the value of strain that is predicted by the neural network is nevertheless significantly greater than the value extracted from experimental topography. This is likely a consequence of subtle differences between the continuum-model calculations and the experimental spectroscopy, which the network attempts to accommodate by including finite strain.

## Discussion

We constructed and demonstrated a ML procedure that can extract the form of the nematic order parameter in TDBG from LDOS data. The key ingredient was the use of several channels that capture the correlations among different energies. Our work has several important implications. First, it shows that the presence and even the strength and internal structure of nematic order can be extracted when the sample exhibits significant heterostrain; this is a crucial aspect for moiré systems where the issue of distinguishing between nematicity and strain has been the subject of debate. Second, our analysis also shows which type of STM data is needed and most useful to extract information about nematicity: as we have seen, the LDOS maps at a single energy, $\mathcal{D}_{\omega_0}(\mathbf{r})$, are not enough to deduce the form of the nematic order parameter and—contrary to what one might have expected—point spectra, i.e., $\mathcal{D}_{r_0}(\omega)$, contain a lot of helpful complementary information for that task (see also the second model discussed in the Supplementary Note 5). Additionally, by studying the influence of inhomogeneous disorder in $\mathcal{D}_{\omega_0}(\mathbf{r})$ maps, we show in Supplementary Note 4 that our ML procedure is highly robust against potential impurities, demonstrating further its generality and ability to disentangle random factors from microscopic physics. We emphasize that this form of solid-state Hamiltonian learning, i.e., of parameterizing the leading terms of a set of microscopic order parameters (like nematic order) or perturbations (such as strain) and extracting their form using multi-channel CNNs can be more broadly applied to other systems—see Supplementary Note 5 where we discuss a toy model for twisted bilayer graphene—and other forms of instabilities, such as the correlated insulators[64,65] or superconductivity. As such, this could open up ways of revealing the form and role of nematic order and other phases for the physics of quantum materials.

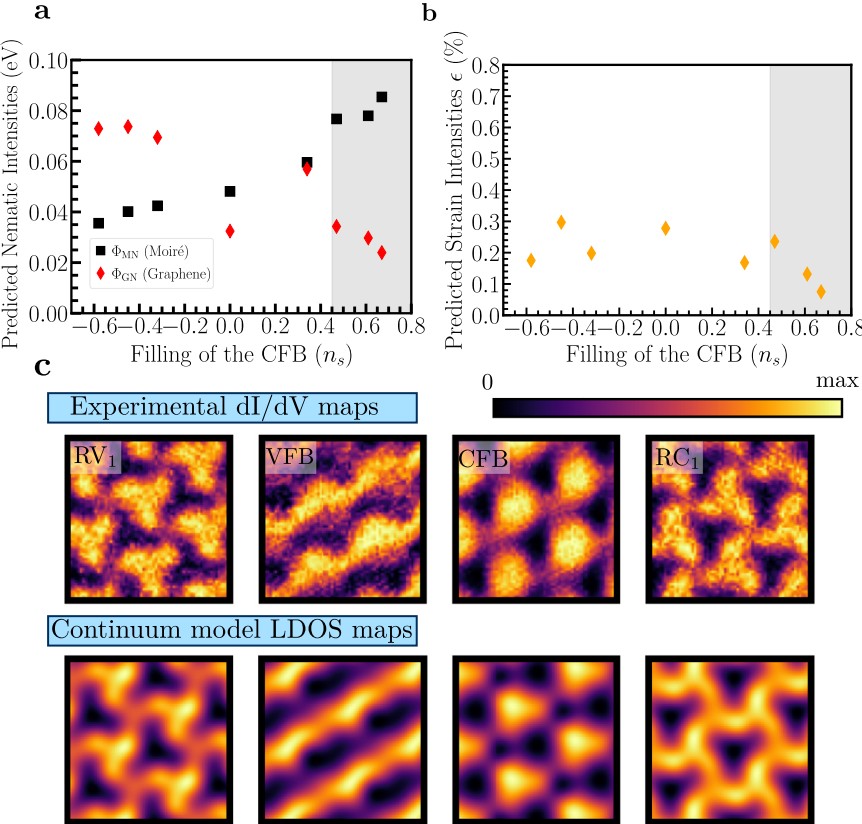

**Fig. 5 | Application to experimental data.** Predicted values from the trained CNN to nematic (**a**) and strain (**b**) intensities as a function of the filling of the CFB ($n_s$). The gray region ($n_s \geq 0.47$) indicates the fillings where the continuum model showed more resemblance to the experimental data obtained in Rubio-Verdú et al.[42]. In panel **c** the experimental $\mathcal{D}_{\omega_0}(\mathbf{r})$ channels for $n_s = 0.67$ are shown for comparison with the ones obtained from the continuum model with the parameters $\beta_{\exp} = \{\Phi_{MN}, \Phi_{GN}, \epsilon\} = \{0.086\,\mathrm{eV}, 0.024\,\mathrm{eV}, 0.05\%\}$ predicted by the trained CNN.

## Methods

### Details on the ML architecture

The implementation of the ML architecture for Fig. 2a was done with the TensorFlow library[66]. Each convolutional layer is followed by batch normalization and max pooling layers (Conv-Batch-MaxPool). The batch normalization layers normalize the input weights in each stage, and also reduce the number of epochs necessary for convergence[67]. This process is repeated four times, with the convolutional layers having a kernel size of $3 \times 3$ and strides set to 1. The filters follow a sequence of 16−32−32−16 with rectified linear unit (ReLU) activation functions[68]. Padding is set to zero such that the reduction of dimensionality is performed only by the MaxPool layers. In turn, these have both strides and pool sizes set to $2 \times 2$. After a Flatten stage, dense layers lead to a dropout before the final layer with filters equal to the number of parameters in $\beta$. The Flatten layer transforms the data to a one-dimensional shape, and the Dropout reduces overfitting by setting a percentage of 20% adjusted weights to zero[69]. Tests on variations of this architecture and the influence of its components on the performance of the predictions are described in Supplementary Note 2.

### Metrics for parity plots

The additional metrics $R^2$ and mean absolute percentage error (MAPE) were calculated via $R^2 = \sum_j^N (\beta_j^{\text{predicted}} - \bar{\beta}^{\text{true}})^2 / (\beta_j^{\text{true}} - \bar{\beta}^{\text{true}})^2 / N$ and $\text{MAPE} = \sum_j^N \left| (\beta_j^{\text{true}} - \beta_j^{\text{predicted}}) / \beta_j^{\text{true}} \right| / N$, where $N$ stands for the number of samples in the test data set, and $\bar{\beta}^{\text{true}} = \sum_j^N \beta_j^{\text{true}} / N$ is the standard mean over the parameters being learned.

## Data availability

The theoretical and experimental data sets used and generated in this study are available in the Zenodo database under the accession code https://zenodo.org/record/7698738.

## Code availability

The source codes used in this study are available in our Github repository[70].

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

## Acknowledgements
J.A.S. and M.S.S. acknowledge funding by the European Union (ERC-2021-STG, Project 101040651—SuperCorr). Views and opinions expressed are however those of the authors only and do not necessarily reflect those of the European Union or the European Research Council Executive Agency. Neither the European Union nor the granting authority can be held responsible for them. This publication was funded by the German Research Foundation (DFG) grant "Open Access Publication Funding / 2023-2024 / University of Stuttgart" (512689491). Salary support is also provided by the National Science Foundation via grant DMR-2004691 (S.T.) and by the Office of Basic Energy Sciences, Materials Sciences, and Engineering Division, U.S. Department of Energy under Contract No. DE-SC0012704 (A.N.P.). J.A.S. is grateful for discussions with J.P. Valeriano, Sayan Banerjee, Patrick Wilhelm, Igor Reis, and Pedro H.P. Cintra. M.S.S. also thanks R. Samajdar, R. Fernandes, and J. Venderbos for a previous collaboration on nematic order in TDBG[46].

## Author contributions
ML and continuum-model calculations for TDBG were performed by J.A.S., S.T., and M.S.S. Analogous calculations for the minimal model of TBG in the SI were done by S.O.; The experimental data set for TDBG from STM measurements was obtained by S.T and A.N.P.; Preprocessing of the experimental data was done by J.A.S. and S.T.; M.S.S. planned and supervised the project. All authors contributed to the writing of the manuscript.

## Funding

## Competing interests
The authors declare no competing interests.
