## [Peer Review file · Nature Communications]

REVIEWER COMMENTS

Reviewer #1 (Remarks to the Author):

This work is about an ML procedure that can extract the form of the nematic order parameter in TDBG from LDOS data. There are several channels to capture the correlations among different energies. This work is also supported by experimental results. The whole manuscript is well written and well represented.

In general this is a demonstration of a novel method, instead of novel phenomena and the discovered properties are regular, so it may not be suitable for publishing in Nature Communications. But it could be publishable in another high-impact journal.

Reviewer #2 (Remarks to the Author):

This study presents a novel machine learning (ML) approach to extract the form of the nematic order parameter in twisted double bilayer graphene (TDBG) from local density of states (LDOS) data. The authors use a convolutional neural network (CNN) to learn the mapping between the LDOS data and the nematic order parameter, which allows for fast and accurate extraction of the latter. The results obtained with the ML approach are compared with reference data, and the good agreement between the two confirms the validity of the method. Overall, this study represents an important step forward in using AI for condensed matter images and for our understanding of the nematic order in TDBG, with implications for the control of quantum phases of matter.

One of the study's strengths is the use of a powerful ML approach to extract the nematic order parameter in TDBG, especially distinguishing nematic order from heterostrain. The authors show that this approach is fast and accurate, and could therefore be of great practical value. One potential weakness of this study is that the authors do not provide a detailed discussion of the limitations of their approach. While the CNN method is shown to be effective for extracting the nematic order parameter from LDOS data, the authors do not quantitatively discuss the potential impact of noise or experimental uncertainties on the accuracy of their method.

Overall, the study presents a step forward in the understanding of these systems, the methodology is adequate, and the results support their conclusions. Therefore, I can recommend publication after major points noted below are clarified.

1. Can you provide more details about the training process, loss curves, number of epochs, computational cost (and hardware used), training times, etc? Also, additional error metrics such as the mean absolute percentage error and R2 for the parity plots.
2. Are the training and results obtained robust on different runs? If you train let's say 10 runs, all of them converge and give the same final results, what is the standard deviation?
3. Related to previous points, without more in-depth error metrics it's hard to assess for overfitting beyond noise. Can you show if and how the results are affected by regularization in the machine learning procedure?
4. Do you have a systematic procedure for determining the number of channels needed, and what energy values are optimal?
5. Was there any investigation into the sensitivity of the extracted nematic order parameters to the choice of hyperparameters in the machine learning procedure?
6. Were there any systematic errors or uncertainties in the LDOS measurements that could affect the accuracy of the extracted nematic order parameters?
7. What is the effect of inhomogeneous disorder in the TDBG sample on the accuracy of the machine learning approach?
8. What are the specific details at which the experimental measurements were taken?
9. Is there a clear significance of the observed correlation between the nematic order parameter and the interlayer potential asymmetry in TDBG?
10. Can you comment on the possibilities to extract additional information about the electronic properties of TDBG beyond the nematic order parameter from the LDOS data using machine learning techniques?
11. What are the potential future directions for research in this field based on the findings of this study? What are the potential challenges in applying this method more generally for experimental images of different systems under different conditions?

Response to Referees for NCOMMS-23-09515:

Machine Learning Microscopic Form of Nematic Order in twisted double-bilayer graphene

Point-by-point response to referee A:

Referee: *This work is about an ML procedure that can extract the form of the nematic order parameter in TDBG from LDOS data. There are several channels to capture the correlations among different energies. This work is also supported by experimental results. The whole manuscript is well written and well represented. In general this is a demonstration of a novel method, instead of novel phenomena and the discovered properties are regular, so it may not be suitable for publishing in Nature Communications. But it could be publishable in another high-impact journal.*

Response: We thank the referee for reading our paper, for deeming it well-written, and for considering it publishable in a high-impact journal. As opposed to the referee, however, we are confident that our work is suitable for Nature Communications, as we explain next:

Firstly, we point out that the study of nematicity in graphene moiré systems constitutes a very timely topic [for instance, note Rubio-Verdú *et al.*, Nature Physics **18**, 196 was published in 2022] and is a very active field of research, as nematic order is believed to be present in a large part of the phase diagram of these (and other) systems and, as such, likely essential to understanding their microscopic physics. However, there are still many open questions such as the precise form of nematicity and whether it can be distinguished from strain. Our work adds significantly to this line of research by providing a way to systematically extract the low-energy form of the nematic order parameter and by showing that it can be distinguished from strain—in fact more than that: even the form and strength of strain can be extracted at the same time. Finally, our work shows what kind of STM data (information) is needed for this task. The main observation is that correlations across different energies are important and that—somewhat counter-intuitively—point spectra at a few high-symmetry points contain more crucial information than full spatial scans at a single energy.

Secondly, we are happy to read that the referee agrees with our view that we introduce and demonstrate “a novel method”. We here, however, would like to emphasize that this novel method is very general and can, thus, be applied to a variety of different physical phenomena (in light of recent STM experiments on the correlated insulator [arXiv:2303.00024, arXiv:2304.10586], extracting its order parameter from data seems to be a very natural application; furthermore, determining the superconducting order parameter from,

e.g., quasi-particle interference, would be another natural opportunity) and other physical systems. Besides mentioning these explicit future applications in the revised version of the manuscript, we have also performed an additional systematic study of the stability of the approach against inhomogeneous disorder [cf. Supplementary Method 4 and Fig. R5 below], revealing that it can disentangle random perturbations from the microscopic physics. This further illustrates the potential and generality of our approach.

In summary, we propose and demonstrate a ML procedure that can systematically extract microscopic physics in the form of parameters of an effective theory of a correlated moiré superlattice from STM data. Applying it to nematicity in TDBG, we show that the nature of nematicity depends on filling and answer the important question whether it can be distinguished from strain affirmatively. In combination with the fact that our methodology can be applied to a great variety of instabilities and systems, the work will appeal to a broad readership, very likely lead to numerous follow-up studies and, thus, seems very much suitable for Nature Communications.

Point-by-point response to referee B:

Referee: *This study presents a novel machine learning (ML) approach to extract the form of the nematic order parameter in twisted double bilayer graphene (TDBG) from local density of states (LDOS) data. The authors use a convolutional neural network (CNN) to learn the mapping between the LDOS data and the nematic order parameter, which allows for fast and accurate extraction of the latter. The results obtained with the ML approach are compared with reference data, and the good agreement between the two confirms the validity of the method. Overall, this study represents an important step forward in using AI for condensed matter images and for our understanding of the nematic order in TDBG, with implications for the control of quantum phases of matter. One of the study's strengths is the use of a powerful ML approach to extract the nematic order parameter in TDBG, especially distinguishing nematic order from heterostrain. The authors show that this approach is fast and accurate, and could therefore be of great practical value. One potential weakness of this study is that the authors do not provide a detailed discussion of the limitations of their approach. While the CNN method is shown to be effective for extracting the nematic order parameter from LDOS data, the authors do not quantitatively discuss the potential impact of noise or experimental uncertainties on the accuracy of their method. Overall, the study presents a step forward in the understanding of these systems, the methodology is adequate, and the results support their conclusions. Therefore, I can recommend publication after major points noted below are clarified.*

Response: We thank the referee for the very detailed summary of our work and for considering it suitable for publication in Nature Communications. We are also thankful for the great comments and questions, which we address in our response below and in the revised manuscript; we feel that this has improved the quality of our manuscript. In particular, this includes an additional study that we conducted motivated by the referee's comments which shows that our approach still works in the presence of additional inhomogeneous disorder (see response to question 7 and Supplementary Method 5 in the revised manuscript).

Referee: *1. Can you provide more details about the training process, loss curves, number of epochs, computational cost (and hardware used), training times, etc? Also, additional error metrics such as the mean absolute percentage error and R2 for the parity plots.*

Response: After creating the dataset for the specific task (defined by learning the set of parameters β), we train the models with a Mean Squared Error (MSE) loss, mean absolute error (MAE) accuracy, ADAM optimizer with a learning rate of 10^{-4} , and a batch size of 64. The training procedure is accompanied by two modules of Keras (i) ModelCheckpoint (https://keras.io/api/callbacks/model_checkpoint/) and (ii) EarlyStopping (https://keras.io/api/callbacks/early_

stopping/): the first is responsible for tracking the validation loss and saving the model at the end of each epoch that corresponds to the minimum loss. The EarlyStopping module will interrupt training when it measures no progress on the validation set (recorded by (i)) after a certain number of epochs. This is defined as a patience argument, usually set around $n_{\text{pat}} = 50$. With this, by setting a total number of epochs of 2000, training usually stops around the 400th epoch. A typical loss curve during the training stage can be seen in Fig. R1. We have observed in all cases a flattening of the loss curve for training and validation close to the same value after typically 300 epochs. We used the set of clusters from the University of Innsbruck (UIBK) (<https://www.uibk.ac.at/th-physik/howto/hpc/regulus.html>) for (i) producing the datasets and (ii) training the ML models. For each run, we typically reserve 40GB of RAM. The specific CPU from the clusters depends on their availability. Although we did not test the computational cost systematically, we also ran the codes on two personal computers A and B which can represent low and high-end performing cases with the following hardware:

- Computer A: (CPU) AMD Ryzen 7 5800H with Radeon Graphics (16) @ 3.200GHz, (GPU 1): AMD ATI Radeon Vega Series / Radeon Vega Mobile Series, (GPU 2): NVIDIA GeForce RTX 3060 Mobile / Max-Q, 16 GB of RAM.
- Computer B: (CPU) Intel i5-7200U (4) @ 2500GHz, (GPU) Intel HD Graphics 620, 12GB of RAM.

For instance, training the model we show the loss curve of in Fig. R1 under the same conditions takes

FIG. R1. Loss curve (mean squared error) as a function of the number of epochs for both training (red lines) and validation (blue lines) datasets for the model with parameters $\beta = \{\Phi_{\text{MN}}, \Phi_{\text{GN}}, \epsilon\}$ from Supplementary Fig. 2 in the SI. Similar loss curves were found for all other sections.

around 2, 6, and 9 hours on the cluster, computers A and B with the adoption of the EarlyStopping module (with $n_{\text{pat}} = 50$). As the referee mentioned, this is a fast and accurate method for the predictions even on older low-end hardware computers such as B. The typical computational bottleneck we found lies in generating the datasets since the continuum model diagonalization for moiré systems can be computationally demanding. Producing 1000 samples with both $\mathcal{D}_{\mathbf{r}_0}(\omega)$ and $\mathcal{D}_{\omega_0}(\mathbf{r})$ channels can take around 12 hours in a setup similar to computer B. Consequently, creating any dataset with 12000 samples from the main sections would take up to one week. In a parallelized environment such as the set of clusters from UIBK, we could generate these datasets reliably in around 13 hours.

All parity plots in the main text and Supplementary Information (SI) now have R^2 and mean absolute percentage error (MAPE) metrics in each respective figure. We also added further details on the training process to the Supplementary Method 4 on the SI. We thank the referee for these suggestions.

Referee: 2. *Are the training and results obtained robust on different runs? If you train let's say 10 runs, all of them converge and give the same final results, what is the standard deviation?*

Response: Yes, they are. We have tested this for all sections and show, as an example, predictions for $\beta = \{\Phi_{\text{MN}}, \Phi_{\text{GN}}, \epsilon\}$ of the main text averaged over $N = 10$ different training runs in Fig. R2a (the architecture and hyperparameters are the same as in Supplementary Method 2 in the SI). Since we estimate the average μ from the samples, the standard deviations are calculated via $\sigma = \sqrt{\sum_i^N (x_i - \mu)^2 / (N - 1)}$, and their corresponding distribution can be seen in Fig. R2b. As suggested, we also present additional error metrics such as R^2 and MAPE which evidence a very good fit between the prediction and true values. We thank the referee for the question and have added a sentence to Supplementary Method 2 stating that the performance of the model is robust on different runs.

Referee: 3. *Related to previous points, without more in-depth error metrics it's hard to asses for overfitting beyond noise. Can you show if and how the results are affected by regularization in the machine learning procedure?*

Response: To exemplify the effect of regularization in the CNN we tested combinations of Dropout and Batch Normalization layers, as well as L1, L2, and L1L2 regularizers for kernel, bias, and activation weights on the Convolutional Layers. For concreteness, we considered the same dataset from Supplementary Method 2 of the SI for these studies (question 1). Among possible combinations of these regularizers, we have observed a more significant influence on the loss curves by having Batch normalization layers before each MaxPool layer, and a final Dropout layer before the final linear layer for the predictions of parameters β . The dropout ratio is usually set to 20%. From the loss curves in Fig. R3, it can be seen that the

FIG. R2. **a** Parity plots for averaged GN, MN, and strain intensities for fixed strain angle $\theta_\epsilon = 0$, with R² and MAPE tests. Each predicted value results from an average of 10 different training runs. Colorbars indicate the corresponding standard deviation for each respective case. **b** Distributions of the standard deviations for each of the three parameters; the maximum of the standard deviation is about one to two orders of magnitude smaller than their typical mean value, indicating that the results are robust on different runs.

gap after convergence of losses between validation and training curves is reduced with the addition of the regularization layers. As such, our choice of batch normalization and dropout was necessary and effective in avoiding overfitting. All results shown in the main text use this form of normalization. We mention this more explicitly in the Supplement of the revised version of the manuscript along with Fig. R3. The combination of L1, L2 and L1L2 regularizers for bias parameters did not reduce overfitting as efficiently, and for activation and kernel regularizers in the convolution layers we observed underfitting with a regularization factor of 0.01.

Referee: 4. Do you have a systematic procedure for determining the number of channels needed, and what energy values are optimal?

Response: Naturally, too few channels simply do not contain enough information to allow for an accurate prediction of the microscopic parameters (e.g., using only a single $\mathcal{D}_{\omega_0}(\mathbf{r})$ was not enough to predict the form of nematicity in Fig. 3 of the main text). While increasing the number of channels of course provides more information and, hence, makes accurate predictions easier in principle, this increases the size of the network and the required (experimental) input data.

Importantly, the “optimal” number of channels strongly depends on the task (what do we want to pre-

FIG. R3. Loss curves (mean squared error) as a function of the number of epochs for ML architectures with (blue line) and without (red line) layers for regularization. The plot was generated with the dataset defined by the parameters $\beta = \{\Phi_{\text{MN}}, \Phi_{\text{GN}}, \epsilon\}$ from Supplementary Method 2 in the SI.

dict?) and also the error tolerance. For each section in the main text, we typically repeated the training process with different numbers and types of channels to investigate their influence on the accuracy of predictions. To illustrate this, we consider the dataset from Fig. 3 of the main text where we want to predict $\beta = \{\alpha, \Phi_{\text{MN}}, \Phi_{\text{GN}}\}$. The validation loss curves are shown in Fig. R4. In this case, the performance with 4 $\mathcal{D}_{\omega_0}(\mathbf{r})$ channels does not get surpassed by the addition of 3 $\mathcal{D}_{\mathbf{r}_0}(\omega)$ channels. When distinguishing strain and nematicity we have found that the inclusion of 7 channels is essential for more accurate predictions on both the theoretical and experimental datasets.

The optimal energy values ω_0 for $\mathcal{D}_{\omega_0}(\mathbf{r})$ maps are motivated by the energy ranges where experimental samples showed stronger signals of anisotropy [Rubio-Verdú, C., Turkel, S., Song, Y. et al. Nat. Phys. 18, 196–202 (2022)].

Referee: 5. Was there any investigation into the sensitivity of the extracted nematic order parameters to the choice of hyperparameters in the machine learning procedure?

Response: Yes, we investigated the performance of the CNN with respect to hyperparameter optimization. These included using different activation functions (SELU, ELU, LeakyReLU, PReLU, ReLU, and Sigmoid), batch sizes (9, 16, 32, 48, and 96), different numbers of filters and convolution layers in the Conv-Batch-MaxPool channels, different learning rates (10^{-1} , 10^{-2} , 10^{-3} and 10^{-4}) and optimizers (RMSprop, SGD and ADAM). The architecture described in Fig. 2a of the main text and its variation in

FIG. R4. Validation loss curves (mean squared error) as a function of the number of epochs for different combinations of types and numbers of DOS channels for predicting $\beta = \{\Phi_{\text{MN}}, \Phi_{\text{GN}}, \alpha\}$.

Fig. 3a already correspond to the optimal configuration we found. Choosing ReLU as activation functions, setting padding to zero in the convolution layers, and using a learning rate of 10^{-4} in the ADAM optimizer had the most significant influence on the accuracy of the predictions.

Referee: 6. *Were there any systematic errors or uncertainties in the LDOS measurements that could affect the accuracy of the extracted nematic order parameters?*

Response: To the best of our knowledge, there are no crucial systematic errors in the experiment. Commonly discussed causes of systematic errors in this type of study are (1) experimental drift, causing the images to become less symmetric, and (2) an asymmetric tip shape. We can rule out (1) because this would impact the positions of the BAAC sites in our images, and we can measure these to be within error of 0% heterostrain. We also checked for this kind of drift carefully while measuring by taking fast and slow images and comparing. An easy way to rule out (2) in our case is simply to note that tip artifacts typically appear on the atomic length scale (the size of the tip apex), whereas the nematicity that we are imaging is on the moiré length scale. Tip effects at the moiré scale would likely be negligible. Also, there are clearly fillings for which the LDOS retains C_3 symmetry, whereas tip artifacts would impact all fillings equally.

From the perspective of the ML approach, we also emphasize that the predicted behavior for the β parameters as a function of the filling seems to be fairly robust with homogeneous (see Supplementary Fig. 3) and inhomogeneous (see discussion on the following question) variations of the experimental $\mathcal{D}_{\omega_0}(\mathbf{r})$ maps in the dataset - if the "disorder" is not too strong, which was typically observed in the experimental samples.

In future studies, comparing the extracted parameters in scenarios where for example, the resolution in the LDOS measurements is increased could give us a better understanding of this aspect.

Referee: 7. *What is the effect of inhomogeneous disorder in the TDBG sample on the accuracy of the machine learning approach?*

Response: As suggested by the referee we investigated the accuracy of the ML approach with the influence of inhomogeneous disorder in the $\mathcal{D}_{\omega_0}(\mathbf{r})$ samples. As such, we consider the same dataset from Supplementary Fig. 2, but now with only 4 $\mathcal{D}_{\omega_0}(\mathbf{r})$ channels.

We simulate data with inhomogeneous disorder using the following procedure: randomly sample the position of a certain number of impurities (N_{imp}) which influence locally the LDOS by a gaussian pixel broadening disorder given by $g(f, \sigma) = f \exp(-r^2/2\sigma^2)/\sqrt{2\pi\sigma^2}$, where f stands for an intensity control. We fix the size of the disorder as 10×10 pixels and $\sigma = 3$. For a given sample (defined by a fixed set of parameters β), the $\mathcal{D}_{\omega_0}(\mathbf{r})$ maps with $\omega_0 = (-35, -15, 1, 23)$ meV have these impurities in the same position for physical consistency. This is repeated for each sample in the entire dataset.

In Fig. R5a we show the R^2 metric for the prediction of the parameters $\beta = \{\Phi_{\text{MN}}, \Phi_{\text{GN}}, \epsilon\}$ as a function of increasing number of impurities (N_{imp}). The red line represents the case without disorder as a reference point. For weak disorder ($f = 30$), we see that even when impurities are present only on the test dataset (dashed green lines) the R^2 metric stays above $R^2 \geq 0.88$ up to $n_{\text{imp}} = 5$. When including the disorder also on the training dataset (solid green lines) we see that the ML procedure seems to learn that the influence of this disorder is not related to the physical parameters β . This realistically describes possible pixel inhomogeneities that could appear in experimental samples. We also show that even predictions on samples with very strong disorder ($f = 100$) can be made more accurately if they are also considered in the training process (solid blue lines). These results contribute to question 6 posed by the referee by suggesting that potential instrumental bias (or systematic errors) can be understood within the ML approach, constituting a particularly interesting scenario also for the study of nematicity beyond moiré systems.

We thank the referee for the great question and added the results of these simulations to the revised version of the manuscript in Supplementary Method 4.

Referee: 8. *What are the specific details at which the experimental measurements were taken?*

Response: The STM measurements were taken at 5K in UHV conditions with a tunnelling set point of 300 mV and 150 pA. Tunneling conductance was measured with a lock-in amplifier with an oscillation amplitude of 1-2 mV. The carrier density was controlled with a voltage applied to a silicon back gate. We mention this more explicitly in the revised version of the Supplementary Method 3.

FIG. R5. **a** R^2 metric as a function of increasing number of impurities (N_{imp}) for predictions of GN, MN and strain intensities. **b** Examples of the influence of this gaussian pixel-disorder in $D_{\omega_0}(\mathbf{r})$ maps are shown for weaker ($f = 30$) and stronger cases ($f = 100$).

Referee: 9. Is there a clear significance of the observed correlation between the nematic order parameter and the interlayer potential asymmetry in TDBG?

Response: Assuming that “potential asymmetry” refers to the displacement field, we have to point out that, unfortunately, STM experiments do not allow for the ideal setup to probe this behavior: since the sample has to be accessible on one side with the STM tip, using gates on both sides is not feasible. Therefore, the filling and the displacement field cannot be tuned independently and one can only change them simultaneously. As such, one cannot formally disentangle the changes in the nematic order parameter coming from displacement field and from electronic filling. As discussed in [R Samajdar *et al*, 2D Materials **8** (3), 034005 (2021)], transport measurements, ideally in a ‘sunbeam’ geometry, might be more useful for this purpose.

Referee: 10. Can you comment on the possibilities to extract additional information about the electronic properties of TDBG beyond the nematic order parameter from the LDOS data using machine learning techniques?

Response: Our studies from both TDBG and TBG (Supplementary Method 5) indicate that as long as any electronic property changes the LDOS maps in a sufficiently non-trivial way, it can be extracted from (sufficiently diverse) LDOS data using a multi-channel CNN setup. Consequently, our work has a general

appeal that goes beyond moiré systems and nematic order, making it a versatile methodology to the study of strongly correlated phenomena. For instance, given that very recent STM experiments [arXiv:2303.00024, arXiv:2304.10586] seem to be able to distinguish between different forms of intervalley coherent order, applying this methodology to extract more details about the microscopic form of the order parameter of the correlated insulators in moiré systems seems to be a promising direction to explore. Similarly, (quasi-particle interference) data on the superconducting state could be used to learn more about superconductivity. We comment on this more explicitly in the revised discussion section and Supplementary Information.

Referee: *11. What are the potential future directions for research in this field based on the findings of this study? What are the potential challenges in applying this method more generally for experimental images of different systems under different conditions?*

Response: There are many exciting future directions for this work both in the context of nematicity in moiré systems and beyond. To mention a few,

- (i) Of course, as already mentioned above (and added to the revised manuscript), the most natural future direction would be to apply the same methodology to other forms of electronic order in twisted graphene systems, such as the correlated insulator [arXiv:2303.00024, arXiv:2304.10586] and possibly superconductivity (likely using quasi-particle interference data).
- (ii) It would be interesting from the perspective of AI to study how transformers-based architectures [Dosovitskiy, A., et al (2020) arXiv:2010.11929.] would perform in similar tasks in comparison to CNNs, given their surprising generality and increasing interpretability [Chefer, H., et al (2021). Proceedings of the IEEE/CVF Conference on Computer Vision and Pattern Recognition.]. The role of each channel of information could be addressed more specifically with the aid of the relevance scores defined in the previously mentioned reference, as an example, to better understand how the ML procedure connects each region on the DOS maps to the physical labels in β .
- (iii) In controlled scenarios (where experimental and theoretical properties for certain phenomena are known) generative adversarial networks (GAN) built experimental STEM datasets could be used to better understand and improve these methodologies [Khan, A. et al, npj Computational Materials 9 (2023)], given the possibility of creating bigger and more realistic training datasets. Additionally, different theoretical models could then be compared to address the shortcomings of the predictions based also on the theoretical limitations to describe the experimental data [Basak, S. et al (2023) Physical Review B, 107(20)].

- (iv) This type of methodology can also offer physical insights from the data that could have been previously missed. In the context of nematicity, a natural following work in TDBG would be the STM high-resolution study of the interplay between graphene nematicity (reminiscent of the untwisted bilayer graphene structures) and moiré nematicity (arising from the moiré structure) as a function of filling of the conduction flat band (CFB) [Cvetkovic, V. et al (2012) Physical Review B, 86(7)].

Concerning general challenges of this methodology when applying to other systems: of course, this depends on the specific system under consideration but an important requirement is that one has a reliable and computationally not too expensive theoretical model for the normal state physics at hand. In particular, in chemically very complex (and/or impure) systems, this might be quite challenging. Furthermore, sufficiently high resolution STM data, containing enough LDOS maps at different energies, is also a prerequisite. However, we are optimistic that modern experimental techniques and computational resources provide the necessary requirements for a successful application of our methodology to a wide variety of systems and phenomena in the near future.

List of changes made

1. Fixed typo in label of Fig. 5 from $\mathcal{D}_{\mathbf{r}_0}(\omega)$ to $\mathcal{D}_{\omega_0}(\mathbf{r})$.
2. Fixed typo in MAE of ϵ for Fig. 4 in main text.
3. Added a new Supplementary Method 4 to discuss points regarding training procedure, limitations of experimental data and robustness of the predictions against samples with inhomogeneous disorder. These are adapted versions of the answers for questions 1, 3, 4 and 7 as suggested by referee B.
4. We explicitly mention applying our methodology to correlated insulators and superconductivity as a promising future direction in the discussion section of the main text and Supplementary Method 4.
5. Added information from experimental measurements in Supplementary Method 3.
6. Parity plots from main and supplementary text now have R^2 and MAPE as additional metrics.
7. Formatting of the main text and SI to the Nature Communications format.

REVIEWERS' COMMENTS

Reviewer #1 (Remarks to the Author):

[Confidential comments to the editor only.]

Reviewer #2 (Remarks to the Author):

I acknowledge the authors for their effective response to the list of comments.

The authors' responses were adequate and the revisions made improved the final version of the manuscript by providing technical validations for many of the computational results.

This was a productive debate, and I now support the revised version for publication.